# Evidence for ethnic inequalities in mortality related to COVID-19 infections: findings from an ecological analysis of England

James Nazroo [ORCID],[1] Laia Becares[2]

[1]Sociology, The University of Manchester, Manchester, UK
[2]Social Work and Social Care, University of Sussex, Falmer, UK

**Correspondence to**
Professor James Nazroo;
james.nazroo@manchester.ac.uk

## ABSTRACT

**Objectives** In the absence of robust direct data on ethnic inequalities in COVID-19-related mortality in the UK, we examine the relationship between ethnic composition of an area and rate of mortality in the area.

**Design** Ecological analysis of COVID-19-related mortality rates occurring by 24 April 2020 and ethnic composition of the population. Account is taken of age, population density, area deprivation and pollution.

**Setting** Local authorities in England.

**Results** For every 1% rise in proportion of the population who are ethnic minority, COVID-19-related deaths increased by 5·12, 95% CI (4·00 to 6·24), per million. This rise is present for each ethnic minority category examined, including the white minority group. The size of this increase is a little reduced in an adjusted model to 4·42, 95% CI (2·24 to 6·60), suggesting that some of the association results from ethnic minority people living in more densely populated, more polluted and more deprived areas.

This estimate suggests that the average England COVID-19-related death rate would rise by 25% in a local authority with two times the average number of ethnic minority people.

**Conclusions** We find clear evidence that rates of COVID-19-related mortality within a local authority increases as the proportion of the population who are ethnic minority increases. We suggest that this is a consequence of social and economic inequalities driven by entrenched structural and institutional racism and racial discrimination. We argue that these factors should be central to any investigation of ethnic inequalities in COVID-19 outcomes.

## Strengths and limitations of this study

► The analysis includes a comprehensive and recent count of deaths, including those occurring outside of clinical settings.
► The situation for different categories of ethnic minority group, including a white minority group, is examined.
► Robust and up to date denominators for the ethnic minority population are used, rather than relying on out of date or inaccurate estimates.
► Partial adjustments have been made for other factors that might explain the association between ethnicity and risk, including those that have been shown to be associated with increased mortality from COVID-19, such as pollution, or that have been found to be risk factors for mortality at the neighbourhood level, such as deprivation and population density.
► However, these analyses are ecological, so inferences cannot be drawn about individuals from these aggregated data and the increased risk of mortality identified may be shared to a certain extent among different ethnic groups in an area.

## INTRODUCTION

There is a growing body of evidence suggesting that there are marked ethnic inequalities in COVID-19-related deaths. Although UK data on ethnicity in relation to COVID-19 are sparse (as are data on ethnicity and health more generally), a report by The Intensive Care National Audit and Research Centre showed that around 35% of COVID-19-related admissions to intensive care were ethnic minority people and ethnic minority admissions were slightly more like to die in critical care (eg, 48·4% of white patients died in critical care compared with 55·3% of ethnic minority patients).[1] Further evidence published by the Guardian and Times newspapers, based on their own, non-peer-reviewed research, suggested that ethnic minority people represent 19% of deaths recorded in hospital,[2] and that areas with a higher proportion of non-white ethnic minority people had higher death rates,[2] while recent analysis of 106 healthcare workers who have died from COVID-19 showed that 63% were from an ethnic minority background, and just over half were not born in the UK.[3] Given that non-white ethnic minority people made up 14% of the population in England and Wales at the 2011 Census, this suggests a marked inequality. This impression was reinforced by secondary analysis of data released by the National Health Service (NHS), which

BMJ

suggested meaningful increases in death rates for ethnic minority people after taking into account differences in age structures and place of residence.[4] Evidence of ethnic/race inequalities in relation to COVID-19 has also been reported in the USA, where regional analyses indicate that areas with larger ethnic minority populations have higher rates of both COVID-19 infections and related deaths.[5] For example, in Michigan, a state where 15% of the population is Black, 40% of deaths are of black people.[6]

Despite this accumulating evidence, it is hard to draw firm conclusions on the extent of ethnic inequalities in COVID-19-related risks. A large proportion of the evidence discussed in media reports is impressionistic, and where statistics on admissions or deaths are collated, denominators are often missing, or are crudely estimated at a national level, rather than estimated from the population from which deaths are counted. In addition, the national data used to calculate denominators are typically drawn from the 2011 Census, which is considerably out of date and almost certainly underestimates the size of the ethnic minority population. Importantly, adjustments are rarely made for the younger age profile of ethnic minority people, nor for the potentially increased exposure to COVID-19 infection that results from their greater concentration in areas with a high population density or greater risk from infection associated with higher levels of pollution and living in areas with greater levels of deprivation. In addition, the data published rarely differentiate between different ethnic groups, and largely ignore white minority groups, meaning that potential differences between different ethnic minority groups are missed. Finally, the count of deaths often only includes those occurring in hospitals, disregarding those who died in community settings.

To begin to address these issues and to enhance our understanding of ethnic inequalities in COVID-19-related risks, we conduct an ecological analysis, at local authority level, of the relationship between COVID-19-related deaths and the proportion of the local population who are ethnic minority. To do this, we use the most recent release of data at the time of writing, accurate estimates of the size and ethnic composition of the population in local authorities, and a modelling approach that accounts for some potential explanations, noted above, for the higher risk faced by ethnic minority people. We note that the approach to explanation is partial, with both incomplete coverage of potential explanatory factors and risk of ecological fallacy.

## METHODS

Death rates are taken from Office for National Statistics (ONS) 5 May 2020 release of deaths in England, by local authority.[7] (Local authorities are government organisations officially responsible for all the public services and facilities for both individuals and business in an area.) This release includes all deaths, including those that occurred in community settings, that occurred up to 24 April and that were registered by the 2 May. Only those deaths recorded as involving COVID-19, a recording that is based on any mention of COVID-19 on the death certificate, are included. These figures do not include deaths of those who reside outside England or deaths where the place of residence is either missing or not yet fully coded on the death certificate. There were 26 004 COVID-19-related deaths in England by 24 April, as recorded on 2 May.

To provide the denominator to calculate COVID-19-related death rates by local authority, the ONS 2018 mid-year population estimates are used.[8] These data estimate the England population as 55 977 178, meaning that the COVID-19-related death rate for England was 46·45 per 100 000 as of 24 April 2020.

To examine the association between COVID-19-related deaths and proportion of the local population who are ethnic minority across local authorities, we use linear regression models with COVID-19-related death rate as the dependent variable. The main predictor variable, the proportion of the population who are ethnic minority, was taken from the ETHPOP estimates for 2018 (we used the 2018 used rather than those for 2020 in order to match in time with the ONS population estimates used for total population size).[9] These estimates are based on sophisticated modelling of change in the ethnic composition of the population at local authority level, accounting for births, deaths and migration both into the UK and within the UK. This is done separately for different ethnic groups, the categories used being: White British/Irish/Traveller; Indian; Pakistani; Bangladeshi; Chinese; Other Asian; Black African; Black Caribbean; Other Black; Mixed; White Other and Other. On 2018, this model estimates that the England population was 77.2% White British/Irish/Traveller. Given the relatively small population sizes of ethnic minority groups and their concentration in particular regions of England, the analysis conducted here combines ethnic minority groups into four categories: Asian (Indian, Pakistani, Bangladeshi, Chinese, Other Asian); Black (Black African, Black Caribbean, Other Black); White Other and Other (Mixed, Other). For the more detailed modelling, all of the ethnic minority groups are combined together and compared with the White British/Irish/Traveller group. In each case, these variables are modelled as a continuous variable that captures a 1% increase in the concentration of the group in a local authority.

The following covariates are included in the analysis: percentage of the population who were aged 70 or over (using the ONS 2018 mid-year population estimates)[8]; population density (number of people per square kilometre, using the ONS 2018 mid-year population estimates)[8]; the 2019 Index of Multiple Deprivation score[10] and levels of pollution as marked by population-weighted annual mean PM V.2.5 (anthropogenic) concentration for 2018 (μgm³).[11]

We use linear regression models to predict variation in COVID-19-related death rate across local authorities. First

**Table 1** Association between size of ethnic population and COVID-19 deaths at local authority level: England

|  | Asian | Black | White minority, excluding Irish and Travellers | Other including mixed | Total ethnic minority |
|---|---|---|---|---|---|
| Death rate per million | 8.94* | 19.85* | 10.30* | 33.04* | 5.12* |
| 95% CI | 6.67 to 11.21 | 15.33 to 24.38 | 6.03 to 14.58 | 25.51 to 40.57 | 4.00 to 6.24 |

*p<0.05.

simple, descriptive models are used to show the association between each ethnic minority category and risk of death and the association between each covariate and risk of death. Then we included all the predictor variables to show the fully adjusted relationship between COVID-19-related deaths and the proportion of the local population who are ethnic minority.

### Patient and public involvement

Patients or the public were not involved in the design, conduct, reporting or dissemination plans of this research. The research was based entirely on secondary analysis of existing administrative data.

### RESULTS

Table 1 presents the results of the crude descriptive models, showing unadjusted coefficients for the relationship between proportion of the population in a local authority who are of an ethnic minority group and the COVID-19-related death rate. For ease of interpretation, the coefficients have been estimated as the change in number of deaths per million for a 1% rise in proportion of the population who are in the ethnic group. Although descriptive in nature, this analysis indicates a marked inequality between areas, with a rise in the ethnic minority population in an area related to a statistically significant higher rate of COVID-19-related death. This is present for each of the ethnic categories shown in the table and also present for each subcategory in a disaggregated analysis (not shown but available from the authors). Most confidence can be placed on the estimate for the total ethnic minority population, which combines all of the other groups because the population size is relatively large and more evenly distributed across local authorities. This coefficient shows that for every 1% rise in proportion of the population who are ethnic minority COVID-19-related deaths increased by 5·12, 95% CI (4·00 to 6·24), per million.

Table 2 presents a descriptive analysis of the relationship between covariates and COVID-19-related death rates in a local authority.

Each of these area characteristics is significantly associated with risk of death, with population density, Index of Multiple Deprivation and level of pollution all increasing risk. However, surprisingly, the proportion of the population who are aged 70 or older, a risk factor for COVID-19-related death, is negatively associated with risk of death, although the effect is very small. Subsequent analysis (not shown, but available from the authors) revealed that this was possibility a result of the negative correlation between the proportion of ethnic minority people and the proportion of people aged 70 and over in an area.

Finally, table 3 presents the full model, with each of the risk factors present. Once all variables are included, only the proportion of the population who are ethnic minority remains statistically significant. The estimated coefficient for proportion ethnic minority is a little reduced compared with the unadjusted figure, going from 5·12, 95% CI (4·00 to 6·24) to 4·42, 95% CI (2·24 to 6·60), suggesting that some of the ethnic minority association results from ethnic minority people living in more densely populated, more polluted and more deprived areas.

To illustrate the significance of these findings, if we take the average England COVID-19-related death rate as 46·45 per hundred thousand, for a local authority with two times the average number of ethnic minority people (using 2018 ETHPOP estimates this would be an increase from 23% to 46%) would have a predicted COVID-19-related death rate of 58·23 per hundred thousand (46·45+(23×5·12/10)), an increase of 25%. While a local authority with a small proportion of ethnic minority people, say 3% rather than 23%, would have a predicted death rate of 36·21 per hundred thousand (46·45−(20×5·12/10)), a decrease of 22%.

**Table 2** Association between area characteristics and COVID-19 deaths at local authority level: England

|  | 1% increase in proportion aged 70 or older | Population density (100 per square km) | Index of multiple deprivation | Pollution |
|---|---|---|---|---|
| Death rate per million | −0.20* | 0.27* | 0.28* | 0.36* |
| 95% CI | −0.26 to −0.15 | 0.19 to 0.35 | 0.084 to 0.56 | 0.25 to 0.47 |

*p<0.05.

**Table 3** Multivariate analysis of association between area characteristics and COVID-19 deaths at local authority level: England

| | 1% increase in proportion ethnic minority | 1% increase in proportion aged 70 or older | Population density (100 per square km) | Index of multiple deprivation | Pollution |
|---|---|---|---|---|---|
| Death rate per million | 4.42* | −0.011 | −0·0023 | 0.098 | 0.10 |
| 95% CI | 2.24 to 6.60 | −0.11 to 0.083 | −0.015 to 0.011 | −0.194 to 0.389 | −0.05 to 0.251 |

*p<0.05.

## DISCUSSION
### Key findings
The analyses presented here provide the strongest evidence to date on ethnic inequalities in COVID-19-related mortality in England. Findings from these analyses show a clear association between an increase in the proportion of ethnic minority residents in a local authority and an increase in mortality related to COVID-19, a finding that is consistent with other literature.[12] The increased risk diminishes slightly, but remains, after controlling for some other characteristics of the area: age profile, population density, level of deprivation and pollution.

### Strengths and limitations
Strengths of this analysis are that it provides robust and up to date denominators for the ethnic minority population, a comprehensive and recent count of deaths examines the situation for different categories of ethnic minority groups, and that it includes adjustments for other factors that might explain the association between ethnicity and risk, including those that have been shown to be associated with increased mortality from COVID-19, such as pollution,[13] or that have been found to be risk factors for mortality at the neighbourhood level, such as deprivation[12 14] and population density,[12] However, it is important to note that these controls are partial and, because of data limitation, only cover some of the relevant risk factors, which we discuss below. Other unmeasured factors may well be relevant to risk of COVID-19-related mortality and may contribute to ethnic inequalities in this risk.

Other limitations to be considered include that these analyses are ecological, so inferences cannot be drawn about individuals from these aggregated data. This means that the increased risk of mortality identified may be shared to a certain extent among different ethnic groups in the area, although the evidence of an association between increased risk and each of the categories of ethnic minority group examined is suggestive of the risk not being equally shared between ethnic minority and ethnic majority people living in an area. In addition, we can draw from the existing evidence documenting ethnic inequalities across a range of socioeconomic, mental health and physical health outcomes,[15–17] including non-COVID-19-related mortality,[18] to support the likelihood that ethnic minority residents suffer from an increased risk of exposure and vulnerability to COVID-19 infection and related mortality, and whose increased risk, therefore, is likely to be a key factor driving these findings.

### Interpretation
There has been much public debate about what might be driving apparent ethnic inequalities in risk of COVID-19-related complications and death. As we have previously described,[19] one existing focus has been on the role of the underlying social and economic inequalities that are experienced by ethnic minority people. These inequalities mean that on average ethnic minority people are more likely to be poorer; have poorly paid and insecure employment; live in over-crowded housing and live in deprived neighbourhoods with high rates of concentrated poverty and increased pollution levels.[20] This, then, results in greater vulnerability to and poorer prognosis from COVID-19 infection. The adjusted model, which shows a reduction in the coefficient related to proportion of the population who are ethnic minority once factors such as these are taken into account, provides some support for these possibilities. However, it is important to note again that these are ecological analyses, so do not account for the very likely possibility that ethnic minority people have poorer circumstances than others living in the same area.

We have also previously described how ethnic minority people are also more likely to be employed in sectors that increase their risk of exposure to COVID-19,[19] such as in transport and delivery jobs, or working as healthcare assistants, hospital cleaners, social care workers and in nursing and medical jobs.[21] Not only do these occupations increase risk of infection[21] but some of these are also occupations that have been the last to receive supplies of personal protective equipment. People in these occupations are now considered to be key workers, but for decades, ethnic minority people working in these jobs have endured job insecurity, low pay and discrimination.

In addition, as noted by ourselves and others,[19] ethnic minority people are more likely to have chronic health conditions that have been linked to increased risk of COVID-19-related complications and mortality such as asthma, diabetes, high blood pressure and coronary heart disease.[22] It is important to note that these health conditions are socially patterned, so the social and economic inequalities faced by ethnic minority people described above lead to their increased prevalence of these health

conditions.[15] As a result, we repeat our previously made suggestion that the increased risks associated with COVID-19 infection faced by ethnic minority people are now a core component of wider ethnic inequalities in health, and these negative consequences are amplified by long established pre-existing ethnic inequalities in health, both of which are driven by social and economic inequalities.[19]

In recognising this complexity, it is also vital to focus on how the social and economic inequalities that are faced by ethnic minority people are driven by entrenched structural and institutional racism and racial discrimination.[23] As we have argued,[19] even though discussion of racism is typically absent from investigations into ethnic inequalities in health, an explanation of ethnic inequalities that stops at social and economic inequalities and does not acknowledge how these inequalities have been, and continue to be, shaped by historical processes underpinned by racism, will be limited in its ability to generate an understanding of, and solutions to, ethnic inequalities. A significant and broad body of research has now documented the role of racism in patterning inequalities in education, employment, income, housing and proximity to pollution.[24–28] In addition, a large number of mental and physical health outcomes, including asthma and hypertension, have been shown to be associated with experiences of racism and discrimination.[17 23 29] Importantly, experiences of racism and discrimination are not one-off, they co-occur and sequentially lead to deepening inequalities in many domains across a person's life course and are transmitted from one generation to the next.[30 31]

### Policy implications and directions for future research

We have previously discussed how excluding racism—the root of ethnic inequalities in COVID-19 infections and related mortality—from scientific and policy discussions around the determinants and implications of the coronavirus pandemic can lead to ineffective investigations and policy interventions.[19] These include unevidenced approaches that question whether ethnic inequalities in COVID-19 may be due to genetic or cultural differences, a line of thinking that risks taking us back into a time of scientific racism, but which is, for example, reflected in a call for research on this issue.[32] We have suggested that before responding to this agenda, we should ask ourselves the simple question: 'What could possibly be the genetic or cultural similarities between an ethnic minority family living in Tower Hamlets, London and another living in Detroit, Michigan, both of whom face an increased risk of COVID-19-related complications and mortality?'.[19] More likely than shared genetic and cultural risks, is that they will both have an increased risk of living in disinvested neighbourhoods with high levels of pollution and concentrated poverty, with insecure and underpaid employment, and in overcrowded conditions with substandard levels of housing. Chances are they have had their lives shaped by institutional and structural racism and have experiences of racial discrimination deeply embedded in their

lives.[15–18 20 23] These are the similarities that policy and research efforts should be paying attention to. Given this, the increased risks faced by ethnic minority people should not have been unexpected, as appears to have been the case, they could and should have been anticipated.

That Public Health England carried out a review of ethnic inequalities in COVID-19-related outcomes could have been an important shift of focus, especially when contemporary policy work around inequalities in health have largely ignored the question of ethnicity.[33 34] However, for that shift to take place, it is crucial that ongoing policy work informing government responses to the coronavirus pandemic considers how current inequalities relate to longstanding ethnic inequalities in health and does not side-step the question of racism, as one government advisor appears to be attempting to do.[35] Similarly, policy development must also focus on the greater harm done to ethnic minority people as a result of government responses to the coronavirus pandemic and, as the cycle of responses develops, we should move quickly to consider how these greater harms might be mitigated. The justification for the pandemic measures is that their estimated effect on reducing the impact of the COVID-19 pandemic on the NHS, by protecting its capacity to provide care for people who become seriously ill as a result of a COVID-19 infection, would offset their acknowledged extremely negative economic, social, health and psychological impacts. That is, the negative is on average judged to be worth the estimated direct health benefits. However, as we have described elsewhere,[19] the situation facing ethnic minority people is far more precarious than 'the average', meaning that these measures are certainly having a more negative effect on ethnic minority people. In addition, some of the more punitive dimensions of 'lockdown', such as changes in the Mental Health Act,[36] police surveillance and discontinuity in the clinical management of pre-existing conditions, are also going to more adversely impact those with racialised identities.

**Acknowledgements** We are grateful for the support provided by Caelainn Barr, Editor, Data projects, Guardian News and Media, and Niamh McIntyre, Niko Kommenda and Antonio Voce, Data projects, Guardian News and Media.

**Contributors** JN and LB were both involved in the design of the study, interpretation of the findings and writing of the manuscript. JN carried out the analysis. The final manuscript was approved by both JN and LB.

**Funding** Nazroo's work on this paper was funded by the Economic and Social Research Council, grant number ES/R009341/1.

**Competing interests** None declared.

**Patient consent for publication** Not required.

**Provenance and peer review** Not commissioned; externally peer reviewed.

**Data availability statement** Data are available in a public, open access repository. The analyses presented here make use of publicly available data. Links to the data are provided within the paper.

**ORCID iD**
James Nazroo http://orcid.org/0000-0001-6744-2207

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
