## [Reviewer comments · BMJ Open]

ARTICLE DETAILS

TITLE (PROVISIONAL)	Evidence for ethnic inequalities in mortality related to COVID-19 infections: Findings from an ecological analysis of England
AUTHORS	Nazroo, James; Becares, Laia

VERSION 1 – REVIEW

REVIEWER	Filipa Sa King's Business School King's College London
REVIEW RETURNED	10-Jul-2020

GENERAL COMMENTS	The paper addresses an important question and is well written. However, I think the regression analysis presented in the paper has a serious limitation - omitted variable bias. There are many factors that affect Covid-19 mortality and that may be correlated with ethnicity. It is not possible to control for all of these factors, which makes it difficult to interpret the results causally. Because of this, it is also difficult to explain why local authorities with a larger ethnic minority population have larger Covid-19 death rates - is it occupation, underlying health conditions, overcrowded accommodation, racial discrimination, or something else? The authors discuss these alternative explanations, but unfortunately the simple regression does not provide an answer to which factors are important. Omitted variable bias is also the likely explanation for the negative coefficient on population over 70, which, although insignificant, is still puzzling. I think the paper is interesting, but am not sure it fits in a medical journal. The discussion about racism is also topical and important, but is not tested in the data and so reads more like an opinion piece.
---

REVIEWER	Isabelle Bray University of the West of England
	I recently published a very similar paper which I think should have been referenced, and makes this less novel: https://authors.elsevier.com/a/1bQKg7bKBu4va
REVIEW RETURNED	21-Jul-2020

GENERAL COMMENTS	This is an appropriate analysis with the usual caveats for an ecological analysis, which are acknowledged by the authors. Strengths include the breakdown by different ethnic groups, including White minority, and the use of recent population estimates by ethnicity. There are some issues that need to be addressed:
---

	 1. The numerical results given in the Abstract are unadjusted - these should be adjusted for confounders. 2. Introduction cites newspaper articles, please instead find the original data used in the article and cite that directly. 3. Text implies that ref 4 is a report by the NHS, but this is not the case. 4. Your choice of confounders needs to be justified by citing the most relevant evidence you can that these are potential confounders (in the Introduction). 5. Table 1 presents results for different ethnic groups, which is really interesting, but these should be adjusted for confounders. 6. It is not clear why the coefficient for 'total ethnic minority' is lower than any of the individual groups which it summarises. 7. The inclusion of age over 70 as a confounder does not make sense if the rates that are the outcome variable are already age-standardised so please check this. It may explain why you got an unexpected result for this variable. 8. Table 3 - CIs are not shown for all estimates, and p-values are introduced which were not in the earlier tables - please be consistent. 9. Towards the end of the first paragraph of the Discussion, ref 12 is covid-related, while refs 13 and 14 are not (although such references do exist in relation to covid). 10. There are other variables that could be mentioned, even if they cannot included in this analysis e.g. physical activity levels, vit D exposure, rates of obesity and diabetes. 11. The fourth paragraph of the Discussion has no references
--	---

VERSION 1 – AUTHOR RESPONSE

Reviewer: 1

The paper addresses an important question and is well written. However, I think the regression analysis presented in the paper has a serious limitation - omitted variable bias. There are many factors that affect Covid-19 mortality and that may be correlated with ethnicity. It is not possible to control for all of these factors, which makes it difficult to interpret the results causally. Because of this, it is also difficult to explain why local authorities with a larger ethnic minority population have larger Covid-19 death rates - is it occupation, underlying health conditions, overcrowded accommodation, racial discrimination, or something else? The authors discuss these alternative explanations, but unfortunately the simple regression does provide answer to which factors are important. Omitted variable bias is also the likely explanation for the negative coefficient on population over 70, which, although insignificant, is still puzzling.

Thank you for this helpful comment. We have noted that adjustments for control/confounding variables is partial and have included a brief discussion on omitted variable bias at the end of the first paragraph of the discussion.

I think the paper is interesting, but am not sure it fits in a medical journal. The discussion about racism is also topical and important, but is not tested in the data and so reads more like an opinion piece.

We believe that it is important for medical audiences to have access to papers that discuss broader explanatory frameworks, including discussion of distal explanatory factors. We appreciate that the

discussion of racism is not tested in the analysis, not least because data on relevant concepts are not available. However, we do reference other research papers to support our discussion of the causal processes that underlie the ethnic inequalities that we document in the paper.

Reviewer: 2

I recently published a very similar paper which I think should have been referenced, and makes this less novel: <https://authors.elsevier.com/a/1bQKq7bKBu4va>

Thank you for alerting us to this paper. We did not previously cite it, because it was not published at the time of the previous version of the paper. We have now included it.

HAVE DOWNLOADED THIS

1. The numerical results given in the Abstract are unadjusted - these should be adjusted for confounders.

Please see our response to the editorial comment no. 1.

2. Introduction cites newspaper articles, please instead find the original data used in the article and cite that directly.

Please see our response to the editorial comment no. 5. Note, that these data were only published in the newspapers cited, and have not been published elsewhere.

3. Text implies that ref 4 is a report by the NHS, but this is not the case.

Thank you, we have modified this text to make it clear that this is secondary analysis of NHS data.

4. Your choice of confounders needs to be justified by citing the most relevant evidence you can that these are potential confounders (in the Introduction).

We had already included a justification for our choice of control variables in the introduction, apologies that this was not clear. We have edited this to be clearer. We also note these at the beginning of the discussion and later we have added a sentence to describe how these only partially control for important explanatory factors.

5. Table 1 presents results for different ethnic groups, which is really interesting, but these should be adjusted for confounders.

We include adjustment for available explanatory factors in Table 3. As described in the text, Table 1 is used to provide a description of the size of the inequality.

6. It is not clear why the coefficient for 'total ethnic minority' is lower than any of the individual groups

which it summarises.

This finding is a consequence of the variations in the geographical distribution and geographical concentration of different ethnic groups. For example, Chinese people are dispersed across a greater number of geographical areas and at a lower level of concentration when compared with Bangladeshi people. The estimates for the total ethnic minority population are, for these reasons, more robust. We describe this in the results section of the paper.

7. The inclusion of age over 70 as a confounder does not make sense if the rates that are the outcome variable are already age-standardised so please check this. It may explain why you got an unexpected result for this variable.

The outcome variable (death rate in a local authority) is not age standardised, it is simply number of deaths divided by population size. So this does not explain the unexpected age result. We note that the age result may relate to the negative correlation between proportion of the population who are ethnic minority and the age profile of the population. This also makes it important to control for age in the analysis.

8. Table 3 - CIs are not shown for all estimates, and p-values are introduced which were not in the earlier tables - please be consistent.

We include confidence intervals only for those estimates that are significant at the $p < 0.05$ level. We have not included p values in earlier tables because all of the coefficients are significant at this level as illustrated by the included confidence intervals, following best practice. For clarity we have not included p values in Tables 1 and 2, but we could add them if reviewers/editors recommend this.

9. Towards the end of the first paragraph of the Discussion, ref 12 is covid-related, while refs 13 and 14 are not (although such references do exist in relation to covid).

We have updated these references and now use COVID related references.

10. There are other variables that could be mentioned, even if they cannot included in this analysis e.g. physical activity levels, vit D exposure, rates of obesity and diabetes.

We included diabetes in our discussion of comorbidities. The contribution of Vitamin D, physical activity and obesity to ethnic inequalities in risk is not clear, so we have not added these.

11. The fourth paragraph of the Discussion has no references,

Apologies, this was an oversight, thank you for spotting it, we have now added the appropriate reference to this paragraph.

VERSION 2 – REVIEW

REVIEWER	Filipa Sa King's Business School King's College London
-----------------	--

REVIEW RETURNED	07-Sep-2020
GENERAL COMMENTS	As I mentioned in my original report, I believe there are two limitations of this study. First, omitted variable bias makes the results less reliable and more difficult to interpret causally. Second, the conclusions are not supported by the empirical analysis and read more like an opinion piece. The revised version makes very limited changes and, in my view, does not appropriately address these two concerns. I think the paper is interesting and well written, but I would not support its publication in the BMJ.
REVIEWER	Isabelle Bray University of the West of England
REVIEW RETURNED	16-Sep-2020
GENERAL COMMENTS	Thank you for addressing my comments. Regarding the issue of confidence intervals in Table 3, I think it is useful to present CIs even when $p > 0.05$, partly because this is an arbitrary cut-off, and the CI gives the reader an idea of the precision of the estimate. I think it looks quite strange to have CIs presented for some results and not others, but I will leave the editor to comment on usual practice for the journal. Thank you for giving me the opportunity to review this paper.

VERSION 2 – AUTHOR RESPONSE

Reviewer: 1

As I mentioned in my original report, I believe there are two limitations of this study. First, omitted variable bias makes the results less reliable and more difficult to interpret causally.

Thank you for this comment. We recognise that in this, and other papers using observational data, it is not possible to control for all of the factors that might be correlated with both the outcome (risk of COVID-19 related mortality) and ethnicity. Similarly, we recognise that this limits our ability to draw strong causal conclusions from the empirical evidence we present. In our previous response to this concern we included a short discussion of this problem in the conclusion of the paper. We have now expanded this in both the introduction to the paper and in the discussion.

At the end of the introduction, just before the methods section and after noting that we account for only some potential explanatory factors, we now include the sentence:

'We note that the approach to explanation is partial, with both incomplete coverage of potential explanatory factors and risk of ecological fallacy.'

The relevant part of the discussion (end of the first paragraph of the strengths and limitations section) now reads:

'However, it is important to note that these controls are partial and, because of data limitation, only cover some of the relevant risk factors, which we discuss below. Other unmeasured factors may well be relevant to risk of COVID-19 related mortality and may contribute to ethnic inequalities in this risk.'

Second, the conclusions are not supported by the empirical analysis and read more like an opinion piece.

As we argued previously, we believe that it is important to place the findings of this paper within the broader academic literature and policy context. To make clear to the reader how we are doing this we have taken on board the Editors' suggestion that we use sub-headings in the discussion. This means that it is clear to the reader where we are summarising the key findings, discussing strengths and limitations of the analysis, interpreting the findings in the context of a broader literature, and discussing implications for policy and future research. Throughout we reference appropriate literature.

The revised version makes very limited changes and, in my view, does not appropriately address these two concerns. I think the paper is interesting and well written, but I would not support its publication in the BMJ.

We hope that this further revision of the paper, and our responses above, address both of the reviewer's concerns. However, we note that there may be some difference between us and the reviewer in terms of the purpose of a paper such as this. As we argued previously, we believe discussing how the findings relate to the broader academic literature and policy context is a crucial additional step. Without this, the reader will not have the relevant information for the interpretation of the significance of findings. This includes a discussion of casual processes and distal explanatory factors that are impossible to empirically capture in this type of paper, but that are evidenced in the broader literature that we cite, such as racism

Reviewer: 2

Regarding the issue of confidence intervals in Table 3, I think it is useful to present CIs even when $p > 0.05$, partly because this is an arbitrary cut-off, and the CI gives the reader an idea of the precision of the estimate. I think it looks quite strange to have CIs presented for some results and not others, but I will leave the editor to comment on usual practice for the journal.

Thank you. The tables have now been edited following this suggestion (and comments from the editors), and now all tables contain confidence intervals, with $p < 0.05$ indicated with an asterisk and footnote.